# Chromosome Doubling Enhances Biomass and Carotenoid Content in *Lycium chinense*

**DOI:** 10.3390/plants13030439

**Published:** 2024-02-02

**Authors:** Runan Zhang, Shupei Rao, Yuchang Wang, Yingzhi Qin, Ken Qin, Jinhuan Chen

**Affiliations:** 1State Key Laboratory of Efficient Production of Forest Resources, National Engineering Research Center of Tree Breeding and Ecological Restoration, College of Biological Sciences and Technology, Beijing Forestry University, Beijing 100083, China; zhangrn2021@163.com (R.Z.); raoshupei1995@163.com (S.R.); y.c.wang420@outlook.com (Y.W.); qyz6138@bjfu.edu.cn (Y.Q.); 2National Wolfberry Engineering Research Center, Ningxia Academy of Agriculture and Forestry Sciences, Yinchuan 750002, China; qinken7@163.com

**Keywords:** colchicine, polyploid induction, tetraploid, morphological features, nutrient composition, *Lycium*

## Abstract

*Lycium chinense*, a type of medicinal and edible plant, is rich in bioactive compounds beneficial to human health. In order to meet the market requirements for the yield and quality of *L. chinense*, polyploid induction is usually an effective way to increase plant biomass and improve the content of bioactive components. This study established the most effective tetraploid induction protocol by assessing various preculture durations, colchicine concentrations, and exposure times. The peak tetraploid induction efficacy, 18.2%, was achieved with a 12-day preculture and 24-h exposure to 50 mg L^–1^ colchicine. Compared to diploids, tetraploids exhibited potentially advantageous characteristics such as larger leaves, more robust stems, and faster growth rates. Physiologically, tetraploids demonstrated increased stomatal size and chloroplast count in stomata but reduced stomatal density. Nutrient analysis revealed a substantial increase in polysaccharides, calcium, iron, and zinc in tetraploid leaves. In addition, seventeen carotenoids were identified in the leaves of *L. chinense*. Compared to the diploid, lutein, *β*-carotene, neoxanthin, violaxanthin, and (E/Z)-phytoene exhibited higher levels in tetraploid strains T39 and T1, with T39 demonstrating a greater accumulation than T1. The findings suggest that the generated tetraploids harbor potential for further exploitation and lay the foundation for the selection and breeding of novel genetic resources of *Lycium*.

## 1. Introduction

*Lycium* is a perennial plant belonging to the Solanaceae family, with seven species and three varieties in China [1]. The three dominant species include *Lycium barbarum*, *Lycium chinense,* and *Lycium ruthenicum*. These species are mostly utilized in food production, medicine, plus soil and water conservation [2]. Wolfberry, also known as goji berry, contains important bioactive substances (such as flavonoids, phenolics, carotenoids, ascorbic acid, thiamine, nicotinic acid, and polysaccharides). These substances bear beneficial effects like blood sugar reduction, liver and kidney nourishment, anti-aging, thirst-quenching, and anti-tumor operations [3,4]. Recently, there has been an increasing inclination towards using natural herbs, particularly those with antioxidant activity, in medicine and food. This trend has significantly boosted the market demand for wolfberry. Consequently, wolfberry has become the primary raw material in the production of drugs, tea, and health products, among others [5]. According to statistics, the wolfberry cultivation area in China expanded to 286,000 hectares in 2023, marking a 10.4% increase from 2022. Simultaneously, wolfberry consumption rose to 78,000 tons, a 15.6% growth compared to the previous year. These statistics indicate a consistent upward trend in the wolfberry industry.

*L. chinense* is extensively cultivated in China and has adaptability and a unique taste. The leaves of *L. chinense* are nutrient-rich, carrying bioactive ingredients and trace elements that are even more beneficial than the fruit [6,7]. Existing research has reported polysaccharides in *L. barbarum* leaves [8] and the quantification of eight phenolic acids and eleven flavonoids in *L. chinense* leaves [9]. *L. barbarum* polysaccharides (LBP) have shown therapeutic promise in treating liver disease, hyperlipidemia, and diabetes [10]. Carotenoids, which serve as precursors and antioxidants for vitamin A synthesis, cannot be synthesized in the body and must, therefore, be obtained through dietary sources such as egg products and green leafy vegetables [11]. The most widely distributed carotenoids mainly include *β*-carotene, lycopene, and lutein [12]. Among them, the antioxidant and anti-inflammatory properties of lutein have shown protective effects against various eye diseases [13]. *β*-carotene has been noted for potentially reducing the prevalence of various chronic diseases and boosting immune system functionality [14]. Lycopene, the most potent antioxidant among all carotenoids, has been shown to protect cells from oxidative damage to lipids, proteins, and DNA [15]. Furthermore, essential minerals present in the leaves play vital roles in cellular metabolism, biosynthesis, and physiological functions [16]. Therefore, the content of these substances is an important indicator for evaluating the nutritional value of *L. chinense*.

Today, the leaves of *L. chinense* are globally consumed, notably in Southern China. Thus, it is necessary to cultivate a new variety with higher yield and nutrient content in a short period of time to meet the market demand. However, this is difficult to achieve using traditional cultivation methods. Polyploidization has been identified as a highly efficient strategy in tree breeding, boasting short breeding cycles and great efficiency. Generally, polyploid plants exhibit rapid growth, large nutritive organs, and high concentrations of secondary metabolites [17], which improve fruit quality, wood production, and the content of active ingredients in medicinal plants [18,19].

Somatic chromosome doubling is recognized as one of the quickest methods of achieving polyploidy [20]. Moreover, polyploidy can also be induced by applying anti-mitotic agents to seeds, stem segments, leaves, somatic embryos, or female gametes [21,22,23], of which colchicine is the most widely used and successful anti-mitotic agent [24]. Technological advancements in in vitro tissue regeneration of plant species have led to the successful development of polyploid variants of numerous plant species, such as grape [25], jujube [26], *Populus* [27], and blueberry [28]. Compared to in vivo methods, in vitro induction improves the efficiency, reduces the occurrence of chimeras, and even allows the isolation of tetraploids from chimeras [28].

Recent years have seen the successful establishment of in vitro tissue regeneration systems using leaves as explants [29,30]. However, there are few reports on the establishment of regeneration systems with *L. chinese* leaves as explants [31,32,33]. In this study, we succeeded in establishing both the in vitro regeneration system and the polyploid induction system using *L. chinense* leaves. This led to the discovery of a new *L. chinense* tetraploid germplasm exhibiting remarkably increased biomass and carotenoid content. Consequently, we established a novel method for the creation of highly nutritious woody food sources.

## 2. Results

### 2.1. Shoot Regeneration from Leaf Explants

Mature leaves were incised with 1–2 cuts and then transferred to MS differentiation medium with different concentrations of growth regulators (Figure 1). After approximately ten days, the callus emerged around the cut area. This callus began to proliferate and accumulate in large amounts in the following 20 days. After 40 days, adventitious shoots started to appear at some leaf incisions. The count of the adventitious shoots regenerated was taken after 50 days.

The appropriate concentration ratios of 1-naphthylacetic acid (NAA) and 6-benzylaminopurine (6-BA) can directly induce the regeneration of adventitious shoots on leaves (Table 1). Concentrations of 6-BA, whether 0.5 mg L^–1^ or 1.0 mg L^–1^, can prompt the direct differentiation of adventitious shoots. However, the induction efficiency at the former concentration surpasses that of the latter. Yet, an increase to 1.5 mg L^–1^ of 6-BA fails to induce such regeneration. Furthermore, the concentration ratio between cytokinin and growth hormone also influences the regeneration of adventitious shoots. The concentration ratio of 6-BA to NAA at 5:1 or 10:1 appears more suitable for shoot regeneration. Following these preliminary results, the experimental design was further developed (treatments 10–15).

The effects of plant growth regulators, 6-BA and NAA, on the regeneration of adventitious shoots from *L. chinense* leaves were evaluated under 15 different treatments (Table 1). It was found that the 6-BA and NAA concentrations significantly influenced both the frequency of shoot formation and the number of adventitious shoots per explant. A lower concentration of 6-BA (0.3 mg L^–1^) appeared more conducive to shoot regeneration, as both the regeneration rate and the number of adventitious shoots regenerated per explant tended to decrease with increased concentration. Furthermore, the optimal results in inducing adventitious shoot regeneration were observed with a cytokinin to growth hormone concentration ratio of 5:1. Overall, a medium containing 0.3 mg L^–1^ of 6-BA and 0.06 mg L^–1^ of NAA proved the most successful in inducing the regeneration of adventitious shoots from leaves.

### 2.2. Effect of Experimental Factors on the Survival Rate of Explants and the Efficiency of Polyploid Induction

A total of 18 treatments were designed to assess the impacts of preculture duration, colchicine concentration, and exposure time on tetraploid induction (Table 2). The survival rates of explants subjected to varying colchicine treatments spanned from 24.1% to 96.1%. This presented significantly lower survival rates in comparison to the control group not exposed to colchicine, attributable to colchicine’s toxic effects. As colchicine concentration increased and exposure time was extended, the survival rate gradually decreased. The lowest survival rate was observed with a 2-day immersion in 60 mg L^−1^ colchicine (<38.9%), under which the regenerated adventitious shoots were frail and displayed poor rooting. Conversely, a 10-day preculture and colchicine treatment at 40 mg L^−1^ for 1 day resulted in the highest explant survival rate.

All three concentrations of colchicine successfully induced tetraploids, but the induction frequency varied across treatments. Lower concentrations of colchicine were less effective in stimulating leaves, leading to reduced tetraploid induction efficiency. Excessively high concentrations and extended exposure times negatively impacted tetraploid induction. Thus, selecting an optimal mutagen concentration was crucial for enhancing leaf regeneration and tetraploid induction efficiency. With a constant preculture duration, the induction efficiency significantly improved (>11.2%) using a 50 mg L^−1^ colchicine concentration with a one-day immersion. The duration of preculture also had a notable impact on induction efficiency. This study found that a 12-day preculture resulted in higher tetraploid induction. In conclusion, the most effective tetraploid induction (18.2%) was achieved with a 12-day preculture followed by a one-day treatment with 50 mg L^−1^ colchicine.

### 2.3. Tetraploid Plants Identification

Using the diploids as a control, the ploidy levels of all regenerated plants were ascertained using flow cytometry. The peak of the control appeared at the 50 channel and the peak of the tetraploids appeared at the 100 channel (Figure 2a,b). Chromosome counts from the stem tips validated the flow cytometry analysis results. Further, cytological analysis revealed that the chromosome number in the diploid plants was 2n = 2x = 24 (Figure 2c) and 2n = 4x = 48 in the tetraploid plants (Figure 2d).

### 2.4. Stomatal and Chloroplast Characteristics Observation

Significant discrepancies were observed in stomatal length, width, and density between diploid and tetraploid plants (Table 3). Tetraploid stomata exhibited a length of 43.08 μm and a width of 35.33 μm; these were 53.15% and 35.52% larger than those of the diploids, respectively. Conversely, diploids displayed a considerably higher stomatal density than tetraploids (Figure 3a,b), with a density 1.72 times higher than that of tetraploids. The mean number of chloroplasts in the tetraploid stomatal guard cells, at 26.73, was significantly higher than that of the diploids (Table 3, Figure 3c,d).

### 2.5. Comparative Analysis of Growth Traits

Both diploid and tetraploid plants were observed for growth traits after 30 days of growth on a rooting medium (Appendix A). There was no significant difference in plant height between the two groups (Figure 4a). However, substantial differences were noted in the growth characteristics of their leaves. Tetraploids showed significantly greater leaf length, width, and area compared to the diploids (Figure 4b). Leaf and stem thickness for both ploidy plants were examined using scanning electron microscopy (Figure 4c–f), revealing that tetraploids visibly possessed thicker leaves and more robust stems than diploids. Following 60 days of growth, both the fresh and dry weights of the whole plants and 10 leaves from the two ploidy plants were assessed. Although tetraploids had a significantly higher fresh weight for both whole plants and leaves, the weight difference was not significant after drying (Appendix A).

Further measurements of plant height, stem thickness, leaf length, width, and area were recorded after a period of 60 days of growth in the greenhouse. The findings indicated that the tetraploid plants, at 31.1 cm, were significantly taller than the diploid plants, with a height of 1.28 times that of the diploids. However, there was no significant difference in stem thickness between the two ploidy plants (Figure 4g, Appendix A). Compared to diploids, leaf length, width, and area were significantly augmented in the lower, middle, and upper parts of tetraploid plants (Figure 4h, Appendix A). Moreover, the tetraploids had visually larger buds with normal flower organ morphology (Figure 4i).

### 2.6. Determination and Analysis of Photosynthetic Pigments in Leaves

Morphological observations indicated that the leaf color of tetraploids was deeper than that of diploids. To compare differences in chloroplast pigment content among the two ploidy plants, the middle leaves from 45-day-old diploid and tetraploid seedlings were analyzed. The data revealed that the content of chlorophyll a, chlorophyll b, and total chlorophyll in tetraploid leaves was significantly higher than those in diploid leaves (Figure 5), accounting for the darker green color of tetraploid leaves. The higher chlorophyll level in tetraploids suggests a higher photosynthetic capacity and activity, enabling the conversion of more photosynthetic products compared to diploids. Additionally, the carotenoid content was also found to be higher in tetraploids than in diploids.

### 2.7. Analysis of the Anatomical Structure of the Leaf

An exploration of the cytological reasons for leaf thickening in tetraploids was conducted by comparing and analyzing the cross-sections of tetraploid and control leaves using paraffin sections (Figure 6). The study revealed that not only were the veins of the tetraploid leaves thickened, but the upper and lower epidermis were also thickened, and the diameter of the xylem cells was significantly increased. Comparatively, the thickness of the palisade tissue and spongy tissue increased by 35.6% and 21.1%, respectively, in contrast to the control. Further, the ratio of palisade tissue to spongy tissue in tetraploids was significantly higher than that of its diploid counterparts (Table 4). These suggested that leaf thickening can be attributed to a notable expansion in cell volume.

### 2.8. Comparative Analysis of Nutrient Content in Leaves

In an attempt to understand variances in nutrient composition, specific constituents, including total sugar, flavonoids, protein, polysaccharides, total amino acids, betaine, and certain minerals (calcium, iron, and zinc) were assessed in the leaves of three tetraploid plants and the diploid control (Figure 7). The results showed that there were dissimilar degrees of differences between T10, T18, T39, and the control. The total sugar content was significantly less in T10 and T39 compared to the control, while T18 recorded no significant difference (Figure 7a). Evaluating the tetraploids as a whole against the diploids, a significant decrease in flavonoid content and an increase in polysaccharides content were observed, while total protein and betaine content remained unchanged (Figure 7b–e). Among the individual samples, T39 exhibited the lowest flavonoid content, whereas T10 had the highest polysaccharide content. Remarkably, all tetraploid strains showed higher mineral content compared to diploid plants (Figure 7g–i), with T39 showcasing the highest iron content and T10 and T18 exhibiting the highest calcium and zinc concentrations. To summarize, tetraploid strains obtained by polyploid induction of diploids differ in their nutrient content.

### 2.9. Analysis of Carotenoid Composition and Content in Leaves

A targeted metabolomics analysis was implemented using HPLC-MS/MS platforms to dissect the divergences in the carotenoid composition and content between diploid and tetraploid strains. The QC analysis revealed a high overlap of TIC curves (Figure 8a), indicating consistent retention time and peak intensity. These findings imply superior signal stability of the mass spectrometry during multiple detections of the same sample, bolstering the reproducibility and reliability of the resulting data. Seventeen carotenoids were identified in the leaves of three tetraploid strains and their control, inclusive of *α*-carotene, *β*-carotene, (E/Z)-phytoene, antheraxanthin dipalmitate, lutein myristate, lutein palmitate, violaxanthin dibutyrate, violaxanthin myristate, antheraxanthin, violaxanthin, neoxanthin, lutein, *β*-cryptoxanthin, 8′-apo-beta-carotenal, *α*-cryptoxanthin, echinenone, and capsanthin (Table 5, Figure 8b). Notably, lutein, *β*-carotene, neoxanthin, and violaxanthin were the major carotenoids in wolfberry leaves. In terms of lutein content, T1 and T39 respectively exhibited 202.2 µg/g and 265.2 µg/g more than the control, while T10 showed a decrease compared to the control. The accumulation of *β*-carotene and neoxanthin increased in T1 and T39 relative to the control, but the variation in T10 was statistically negligible. Violaxanthin accumulation was approximately similar in the control and T1, with both significantly lower levels compared to T10 and T39. The content of (E/Z)-phytoene in T1, T10, and T39 increased by 14.0%, 46.9%, and 41.6%, respectively, compared to the control. In conclusion, the content of carotenoids in the leaves ranked as follows: T39 > T1 > CK > T10.

## 3. Discussion

This is the first report on the direct induction of adventitious shoot regeneration from leaf explants of *L. chinense* and the colchicine treatment of diploid leaf explants to generate tetraploid plants. Plant cells, known for their totipotency, can regenerate whole plants from somatic cells without fertilization [34]. Plant growth regulators play a crucial role in in vitro plant regeneration from various plant tissues, with cytokinin and growth hormones being the most remarkable.

During the regeneration process from explants, maintaining a balanced cytokinin to growth hormone ratio is crucial. A higher cytokinin ratio relative to growth hormone promotes adventitious bud formation. Zhao et al. [35] found that the highest regeneration frequency was obtained for *Brassica rapa* L. on a medium containing 2.0 mg L^–1^ 6-BA and 0.5 mg L^–1^ NAA. Song et al. [36] identified that the optimal medium for “Jinyan” kiwifruit leaf adventitious shoot regeneration was MS + 6-BA 4.0 mg L^–1^ + NAA 0.1 mg L^–1^. However, excessively high concentrations of growth regulators impeded direct shoot organogenesis [37]. In this study, the most efficient adventitious shoot regeneration occurred at a 6-BA concentration of 0.3 mg L^–1^, with 6-BA being five times the concentration of NAA. As the concentration of growth regulators increased, the frequency of adventitious bud regeneration decreased, dropping to zero when both cytokinin and growth hormone concentrations reached 1.5 mg L^–1^. These findings are consistent with prior theories. This research paves the way for establishing a multi-genotyped wolfberry regeneration system. When combined with the polyploid induction system, this approach could yield superior polyploid germplasm, fulfilling the diverse requirements of the market.

Colchicine primarily functions by regulating microtubule stability. Colchicine interacts with microtubule proteins, inhibiting microtubule polymerization and obstructing the formation of a stable cytoskeleton. Consequently, during mid-mitosis, microtubule formation fails, hindering chromosome movement towards the poles, which results in polyploid cell formation [38]. The efficiency of polyploid induction is closely linked to colchicine concentration and soaking time. The polyploidy induction protocols of *Zizyphus jujuba* Mill. cv. Zhanhua and *Zizyphus jujuba* Mill. var. *spinosa*, both of the same genus Ziziphus, had significantly different colchicine concentrations and treatment durations [39,40]. These variations are attributable to the unique characteristics and growth states of different plants, resulting in different sensitivity and tolerance to colchicine, and the induction efficiency of polyploidy also varies. In this study, the optimal polyploid induction efficiency was achieved by immersing explants in 50 mg L^−1^ colchicine for 24 h. At a colchicine concentration of 60 mg L^−1^, the adventitious shoots exhibited weak vigor and severe browning, with most dying before rooting, negatively affecting tetraploid induction. This is probably due to the thin leaves of *L. chinensis*, resulting in low tolerance to colchicine. Additionally, preculture time is critical for tetraploid induction, with different plants requiring various preculture durations. For instance, *Populus hopeiensis* necessitated 9 days [41], while *Lilium davidii* var. *unicolor* required 15 days [42]. In this study, explants were treated with colchicine after 12 days of preculture, which yielded the highest success in tetraploid induction. However, extending the preculture beyond 12 days often resulted in the formation of chimeras. Therefore, in the plant polyploidy processes, it is crucial to control the concentration of mutagens and treatment duration and closely monitor the growth status of the explants to determine the appropriate preculture time.

Flow cytometry [43] and the determination of chromosome number [44] are widely used to identify the ploidy level in plants. Stomatal characteristics and the number of chloroplasts in guard cells also serve as effective methods to ascertain the ploidy level. In this study, ploidy levels were positively correlated with stomatal size and the number of chloroplasts in guard cells while presenting a negative correlation with stomatal density. This has been demonstrated in many plants, such as *Populus* [45] and watermelon [46]. Therefore, both stomatal traits and chloroplast number are significant phenotypic markers for determining polyploidy levels.

Chromosome doubling in the majority of plants typically leads to similar morphological modifications, such as larger and thicker leaves, dark green leaf coloration, robust stems, and shorter internodes [47,48]. The tetraploids induced in this study had different phenotypic changes in addition to larger and thicker leaves, darker leaf color, and thicker stems. Compared to diploids, these tetraploids had lengthier internodes. Although plant height differences were not significant in early growth stages, some tetraploid strains significantly outgrew diploids in later stages, aligning with findings by Noori et al. [49] and Dai et al. [50]. This height disparity may correlate with heightened ploidy levels, as the doubling of DNA in tetraploids can enhance cellular metabolic activities, thereby accelerating growth [51]. The variability in tetraploid phenotypes resulted in increased fresh weight of the aboveground parts and leaves of *L. chinense*, contributing to greater biomass. Leaf anatomy observations in our research indicated that the increased thickness of tetraploid leaves is associated with the expansion of cell volume, which was consistent with the previous findings [52,53]. Additionally, the chlorophyll content in L. chinense tetraploids exceeded that in diploids, accounting for the tetraploid leaves’ darker green color. This increase in chlorophyll content following polyploidization was also noted in *Lilium* [54]. Given its high sensitivity to environmental conditions, the microstructure of the leaf can potentially indicate a plant’s adaptability to environmental conditions [55]. Compared to diploid plants, the tetraploid plants exhibited thicker leaf veins and palisade tissues, a higher ratio of palisade to spongy tissue, and increased water content. The thicker the leaf, the stronger the water storage capacity, and the larger the midvein diameter, the stronger its water control ability. This leads to the hypothesis that tetraploids possess superior tolerance to water stress than diploids [56].

Previous studies have shown that polyploid strains induced from plants of different genotypes or varieties differ in growth characteristics and nutrient content. This study found that tetraploid strains differed in plant height. In order to detect whether there were other differences among tetraploids, the tetraploids were divided into three categories based on height variance. Proteins, fats, minerals, bioactive substances (such as LBP, betaine, carotenoids, phenolics, flavonoids), and essential minerals (including calcium, iron, and zinc) in the leaves were measured. The three tetraploids under investigation displayed significant surges in levels of calcium, iron, zinc, and polysaccharides compared to the diploids. However, variability occurred among the tetraploid samples. T10 exhibited the highest polysaccharide content, while T18 showed the least. T39 carried the highest iron content but demonstrated the lowest levels of calcium and zinc. This study also revealed distinct variations in carotenoid accumulation between different tetraploid strains of *L. chinense* and the control group. Specifically, lutein, *β*-carotene, neoxanthin, violaxanthin, and (E/Z)-phytoene exhibited higher levels in strains T39 and T1 compared to the diploid, with T39 demonstrating greater accumulation than T1. However, the content of other constituents showed negligible changes or even slight decreases in strain T10, aside from violaxanthin and (E/Z)-phytoene, which increased markedly compared to the control. This was similar to previous research findings, five traits showed genotype-specific changes after genome doubling in two genotypes of apples [57], and different genotypes of *Artemisia cina* also demonstrated morphological characteristics and metabolite content variations following polyploidy [58].

A positive correlation between ploidy level and plant metabolite content has been observed in some tetraploid plants induced in vitro. One primary objective of inducing polyploidy is to augment metabolite expression and concentration by increasing the chromosome count, particularly in medicinal plants. Recent advancements have seen various medicinal plants, such as *Melissa officinalis* L. [59] and *Salvia multicaulis* Vahl [60], enhance their medicinal ingredient content through chromosome doubling. Our study revealed significantly higher levels of nutrients and bioactive components in tetraploid plants compared to diploids. These findings align with observed increases in crude protein, ash, calcium, and phosphorus contents in tetraploid *Moringa oleifera* Lam. [61]. Similarly, Mahanta et al. [62] also confirmed that carotenoid content was increased in tetraploid gerbera hybrids compared to diploids, and they suggested that it was mainly driven by polyploidization. However, because of much more complex gene expression and regulation in repetitive genomes, induced polyploids do not always live up to the initial assumptions. Conflicting results have been reported for different polyploid species. For instance, a marked decrease in rosmarinic acid was noted in tetraploid *Salvia officinalis* compared to its diploid counterpart [63], and no significant variances were observed in carotenoid content across different ploidy levels in *Stachys byzantina* L. [19]. These findings indicate that polyploid evolution in plants cannot be explained by a singular hypothesis.

Our research explored the characteristics of *L. chinense* tetraploids, encompassing growth traits, leaf anatomy, and the analysis of nutrients, bioactive compounds, and carotenoid metabolomics. Although progress has been made, we still lack a thorough understanding of polyploidization. The phenotypic variations resulting from chromosome doubling are highly complex and are influenced by more than just single genes or gene classes. Consequently, our subsequent research will analyze the mechanism of polyploid phenotypic variation from transcription, metabolism, protein, and epigenetic aspects. On the other hand, our future research will aim to continuously evaluate the growth characteristics, biomass, metabolite content, and fruit quality of different tetraploid plants through field experiments, contributing to the selection of superior tetraploid germplasm.

## 4. Materials and Methods

### 4.1. Plant Materials

We collected the trial materials in Ningxia, and vegetatively propagated in vitro. Uniformly growing plant stems were selected and segmented into 2-cm sections, each with 1–2 leaves, before being inoculated into a rooting medium (1/2 MS 2.47 g/L + sucrose 30 g/L + agar 7 g/L + IBA 0.3 mg/L, pH 5.8).

### 4.2. Adventitious Shoot Regeneration from Leaf Explants

Fully expanded leaves were harvested from sterile-rooted plantlets. Young leaves at 3–6 leaf positions were selected as explants. Leaf tips and margins were removed, and the leaves were segmented into small pieces roughly 0.5 cm wide. These pieces were then inoculated onto a medium designed to differentiate adventitious buds containing varying concentrations of 6-BA and NAA. In total, nine treatments were conducted, and each was repeated three times with 60 explants per treatment. Cultures were maintained at 24 ± 1 °C under illumination of 30–40 μmol m^−2^ s^−1^ for a 16-h photoperiod. After 40 days of culture, both the regeneration frequency and the number of regenerated adventitious shoots per explant were recorded. Here, the regeneration frequency is the ratio of the number of adventitious shoots formed to the initial total number of explants.

Based on the preliminary screening results, the concentration range of 6-BA required for leaf-induced regeneration of adventitious shoots and the more suitable concentration ratio of 6-BA to NAA can be basically determined, and the concentration and ratio of cytokinin and growth hormone can be further adjusted to optimize the differentiation medium.

### 4.3. Colchicine Treatment for Inducing Polyploidy and Plant Recovery

After 30 days of subculture, fully expanded leaves at the 3–6 leaf positions were selected for polyploid induction. Leaf explants were wounded with two transverse cuts without full separation and then inoculated on an adventitious shoot regeneration medium (a solid MS basal medium containing 3% (*w*/*v*) sucrose, 0.7% (*w*/*v*) agar, 6-BA 0.3 mg L^−1^, and NAA 0.06 mg L^−1^). This medium was used for a period of 10, 12, or 14 days. Afterwards, explants were transferred to the same medium containing filter-sterilized colchicine at a concentration of 40, 50, or 60 mg L^–1^ for 1 or 2 days of treatment in darkness. After colchicine treatment, the leaves were rinsed three times with sterile distilled water and positioned on the original adventitious shoot regeneration medium without colchicine to induce differentiation. The entire experiment was repeated thrice with 20 explants for each treatment. After six weeks of culture, the survival rate of the explants was calculated. When regenerating shoots attained a height of 2 cm, individual adventitious shoots were excised and placed on a half-strength MS medium supplemented with IBA 0.3 mg L^−1^ to promote rooting.

### 4.4. Flow Cytometry Analysis of Plant Ploidy

Flow cytometry ploidy analysis was initiated only when at least three fully expanded leaves had been produced to count the tetraploid induction rate. Once the adventitious shoots regenerated from the isolated leaves had reached a height of 1 cm and had been transferred to a rooting medium for 30 days, 0.1 g of young leaves from the plant to be tested was collected and placed in a 6 cm diameter Petri dish, to which 2 mL of modified Galbraith’s buffer (comprising 45 mM MgCl_2_·6H_2_O, 20 mM MOPS, 30 mM sodium citrate, 0.5% Triton X-100, and 1% PVP-10, with a pH of 7.0) was added [64]. The leaves were rapidly sliced with a unilateral blade before being filtered through a 30 μm nylon mesh into a centrifuge tube. Subsequently, 200 μL of 4′, 6′-diamidino-2-phenylindole (DAPI, 10 μg/mL) was added to the tube and the mixture was stained for 2 min, protecting it from light exposure. Plant ploidy was then detected using Cyflow^®^ Ploidy Analyzer (Partec). The fluorescence intensity peak was calibrated to 50 channels with the diploid as a control, and tetraploid plants identified should have a fluorescence peak at 100 channels.

### 4.5. Chromosome Counting

After 15–20 days of growing on root induction medium, plants of both ploidy levels had approximately 5 mm samples taken from their stem tips between 8 and10 A.M. These samples were pretreated with a 0.2% colchicine solution for 6 h at 4 °C. After this period, samples were washed three times with sterile water and situated in a freshly prepared Carnoy’s solution (comprising a 3:1 ratio of ethanol and acetic acid) for 24 h, also at 4 °C. The fixed stem tips were subsequently transferred into 1 N HCl and dissociated at room temperature for 15 min. They were then washed three times with sterile water, each rinse lasting 8 min, to ensure the complete removal of the HCl. The individual stem tip was transferred to a glass slide and stained with modified phenol fuchsine solution (comprising basic fuchsin, carbolic acid, and acetic acid) for 5 min. The sample was clamped with forceps, covered with a coverslip, and lightly tapped with a pencil or rubber implement to disperse the cells. Cell observation was carried out using an Olympus BX51 microscope under a × 100 oil lens.

### 4.6. Stomatal Characteristics Analysis and Chloroplast Count

To observe stomatal characteristics, samples from the lower epidermis of 3–4 mature leaves, proceeding apically downward from both diploid and tetraploid plants, were extracted. These were positioned on slides with distilled water and observed under a light microscope at 400× magnification. From each leaf, ten fields of view were selected at random for observation of stomatal density. Subsequently, in each field of view, thirty stomata were randomly selected for stomatal size measurement. The number of chloroplasts between two guard cells in each stomatal pore was also observed by laser confocal microscopy (Leica TCS SP8 CARS, Leica Biosystems, Wetzlar, Germany). Ten stomata per leaf were randomly selected and used for chloroplast count.

### 4.7. Characterization of Diploid and Tetraploid Growth Traits in L. chinense

The shoots of diploid and tetraploid plants were inoculated into the rooting medium at the same time. After a duration of 30 days, differences in growth were assessed. Plant height was recorded using a tape measure, while leaf length, width, and area were quantified using Image J 1.53k software. Both ploidy plants’ leaf and stem thickness were examined using scanning electron microscopy (SEM). This involved rapidly fixing the middle of the fifth leaf and the stem segment between the fifth and sixth leaf positions in 2.5% glutaraldehyde for 2 h at room temperature. Post-fixative wash was carried out using distilled water, followed by step-by-step dehydration with increasing concentrations (30%, 50%, 70%, 80%, 90%, 95%, and 100%) of ethanol for 10 min at each concentration. The samples were then placed in tert-butanol for 1–2 h before immediate freeze-drying. Then the transverse sections of leaves and stem segments were observed using SEM (Hitachi S-3400N). The fresh weight (FW) and dry weight (DW) of the whole plant and 10 leaves were measured after 60 days of growth. The plants were grown on a rooting medium for 60 days, then transplanted and grown in a greenhouse for a further 60 days. Growth traits like plant height, stem thickness, leaf length, leaf width, and leaf area were again measured.

### 4.8. Determination of Photosynthetic Pigments

Following a growth period of 45 days, leaves from the mid-section of diploid and tetraploid plants were harvested and sectioned into 2 mm-wide fragments. Then, three 0.1 g samples were drawn from each of the plants with different ploidies and placed in test tubes containing 10 mL of a specially prepared extracting solution (composed of anhydrous ethanol, acetone, and water at a ratio of 4.5:4.5:1). These samples were securely stored in light-protected conditions. After extraction for 32 h at room temperature, the absorbance values at wavelengths of 663 nm, 645 nm, and 470 nm were recorded. The concentrations of chlorophyll a, chlorophyll b, and carotenoids were then calculated using Arnon’s formula.

### 4.9. Anatomical Determination of Leaves

The fifth leaf of both the tetraploid and control plants, aged 30 days, were separately harvested for paraffin sectioning. Each leaf was trimmed into a fragment measuring 0.5 cm in length and 0.3 cm in width, with the middle of the main vein as the center. These pieces were then preserved using Formalin–Acetic Acid–Alcohol (FAA) solution and processed through a standard paraffin sectioning procedure. This involved staining with safranine solution for 2 h, followed by decolorization and subsequent staining with the fast green solution for 1 min. The sections were then examined and imaged under a bright field using an Olympus BX51 microscope (Olympus LS, Tokyo, Japan).

### 4.10. Detection of Nutrient Content in Leaves

Using the diploid as a control, the various tetraploid seedlings were grouped into three categories based on differences in plant height; these were shorter than the diploid, equivalent to the diploid, and taller than the diploid. One sample was selected from each category, namely T10, T18, and T39, for testing and analysis of nutrient content in the leaves. The protein content was assessed using the Kjeldahl nitrogen determination method (GB 5009.5-2016). Determination of flavonoids by plant flavonoid assay kit (Biodee (Beijing) Inc., Beijing, China) The content of polysaccharides was identified using the phenol–sulfuric acid assay (SN/T 4260-2015). Determination of total sugar by 3,5-dinitrosalicylic acid (DNS) method [65]. Determination of betaine content by colorimetry (NY/T 1746-2009). Determination of total amino acids (GB 5009.124-2016). Calcium, iron, and zinc contents determination (GB/T 35871-2018).

### 4.11. Determination of Carotenoid Composition

The tetraploid seedling lines T1, T10, and T39, along with their respective diploid control, were cultivated on a rooting medium for two months. For the evaluation of carotenoid compositions, both upper and middle mature leaves were gathered. Three replicates were taken from each sample. MetWare company (Wuhan, China) performed the carotenoid composition determination. The samples were freeze-dried and pulverized before being subjected to an extraction process using a mixture of hexane, acetone, and ethanol (1:1:1, *v*/*v*/*v*) comprising 0.01% butylated hydroxytoluene (BHT). After vortexing at room temperature for 20 min, the suspension was centrifuged at 12,000 revolutions per minute for 5 min at 4 °C. The supernatant was then collected, and the extraction procedure was reiterated. Upon the concluding extraction, the samples underwent a secondary centrifugation, and the supernatants were combined. The resulting extract was concentrated and re-dissolved in 100 μL of dichloromethane. The solution was then filtered through a 0.22 μm filter membrane and prepared for LC-MS/MS analysis.

The sample extracts were analyzed using a UPLC-APCI-MS/MS system (UPLC, ExionLC™ AD, SCIEX, MA, USA; MS, Applied Biosystems 6500 Triple Quadrupole, SCIEX, MA, USA). The analytical conditions were as follows, LC: column, YMC C30 (3 μm, 100 mm × 2.0 mm i.d); solvent system, methanol: acetonitrile (1:3, *v*/*v*) with 0.01% BHT and 0.1% formic acid (A), methyl tert-butyl ether with 0.01% BHT (B); gradient program, started at 0% B (0–3 min), increased to 70% B (3–5 min), further increased to 95% B (5–9 min), and finally ramped back to 0% B (10–11 min); flow rate, 0.8 mL/min; temperature, 28 °C; injection volume, 2 μL [66].

The linear ion trap (LIT) and triple quadrupole (QQQ) scans were conducted utilizing a QTRAP^®^ 6500+ LC-MS/MS System, a triple quadrupole-linear ion trap mass spectrometer with an APCI Heated Nebulizer. The apparatus was operated in the positive ion mode. The entire process was governed by Analyst 1.6.3 software (Sciex). The APCI source was operated with the following parameters: an ion source, APCI+ source temperature of 350 °C; the curtain gas (CUR) pressure set at 25.0 psi; and the collision gas was medium. The assessment of carotenoids was performed using the scheduled multiple reaction monitoring (MRM) technique. Mass spectrometer parameters, including the declustering potentials (DP) and collision energies (CE) for individual MRM transitions were carried out with further DP and CE optimization. During each period, a particular set of MRM transitions was monitored based on the metabolites that eluted within that time frame [67,68].

### 4.12. Sample Quality Control Analysis

A mixed solution was used as the quality control (QC) sample during the procedure of instrumental analysis. Typically, a QC sample is inserted every ten detection and analysis samples. The stability of the instrument during the sample detection can be judged through overlapping display analysis of the total ion current (TIC) diagram obtained from mass spectrometry detection and analysis of the same QC sample.

### 4.13. Data and Statistical Analysis

Analysis of variance (ANOVA) was performed with SPSS (version 24, IBM, USA). The significance of the differential effects between 6-BA and NAA on adventitious shoot regeneration was evaluated via Duncan’s multiple range tests, observing a probability level of 0.05. The analytical approach was consistently applied to examine the influence of preculture time, colchicine concentration, and exposure time on the rate of tetraploid induction. Measurements of stomatal size, stomatal density, leaf thickness, leaf area, and stem thickness were executed using Image J 1.53k software. Additionally, an independent sample *t*-test was employed to compare the growth traits’ disparities between diploid and tetraploid plants.

## 5. Conclusions

In conclusion, we effectively established the *L. chinense* explant regeneration system and tetraploid induction system. The most efficient induction of tetraploids was realized when leaves were precultured for 12 days and immersed in a colchicine concentration of 50 mg L^–1^ for a period of 24 h. The induced tetraploids exhibited standard polyploid traits, such as larger and thicker leaves, as well as more robust stems. Furthermore, noticeable variations were observed in growth characteristics and nutrient contents, particularly in polysaccharides and carotenoid accumulation among the different tetraploid plants. Overall, polyploidy induction increased the biomass, nutrient and metabolite contents of *L. chinense*, which showed excellent phenotypic variation.

## Figures and Tables

**Figure 1 plants-13-00439-f001:**
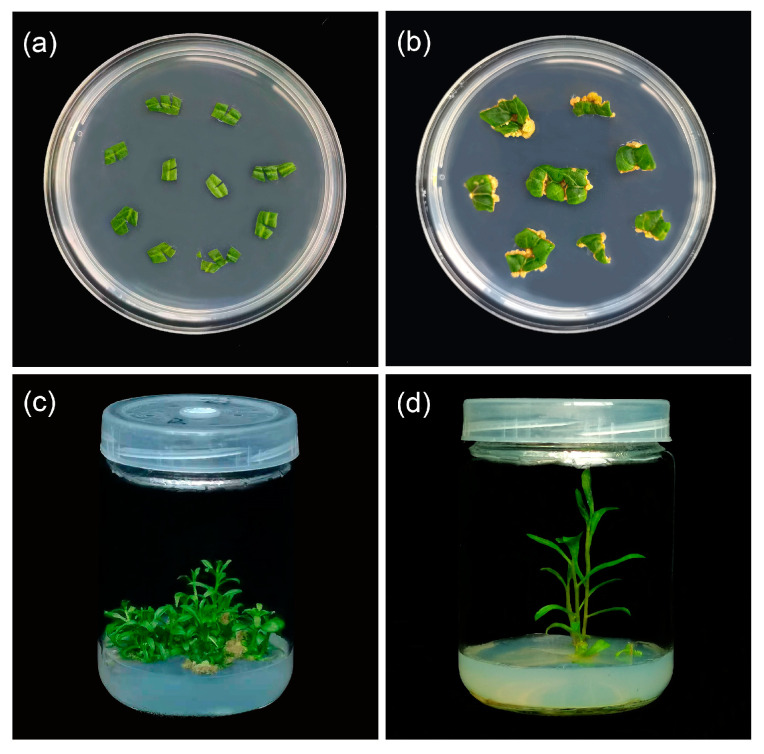
Plant regeneration from leaf explants of *L. chinense*: (**a**) the leaves were transferred to MS differentiation medium with different concentrations of growth regulators; (**b**) a callus has appeared at the incision; (**c**) the leaves successfully regenerated adventitious shoots after 50 days; (**d**) the adventitious buds have rooted.

**Figure 2 plants-13-00439-f002:**
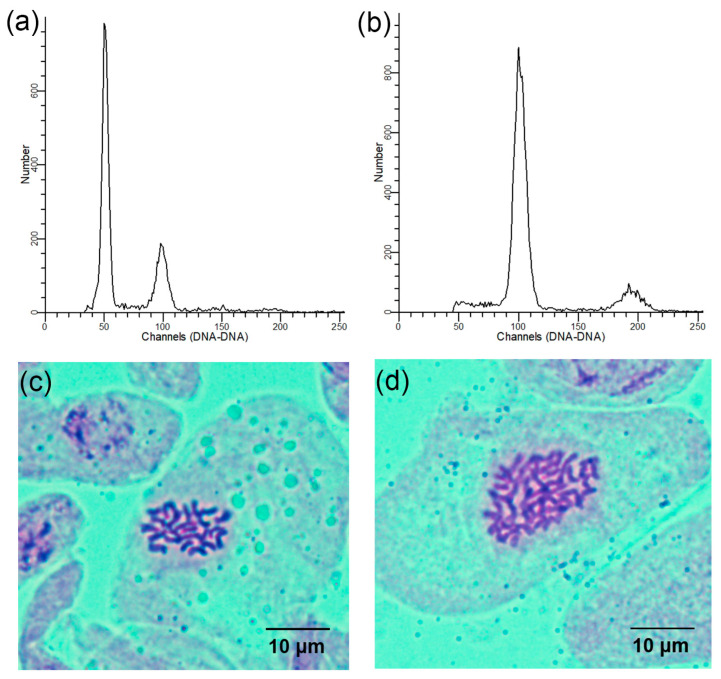
Determination of the ploidy levels in the regenerated *L. chinense* plants: (**a**) flow cytometry histogram of diploids; (**b**) flow cytometry histogram of tetraploids; (**c**) somatic chromosome number of diploids (2n = 2x = 24); (**d**) somatic chromosome number of tetraploids (2n = 4x = 48).

**Figure 3 plants-13-00439-f003:**
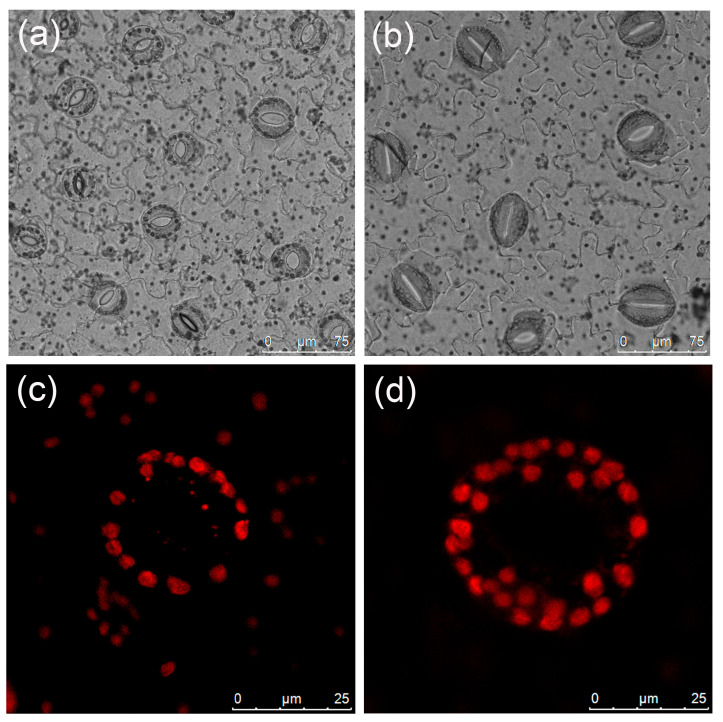
Stomatal and chloroplast characteristics of *L. chinense*: (**a**) diploid stomata; (**b**) tetraploid stomata; (**c**) chloroplasts in the diploid guard cell; (**d**) chloroplasts in the tetraploid guard cell.

**Figure 4 plants-13-00439-f004:**
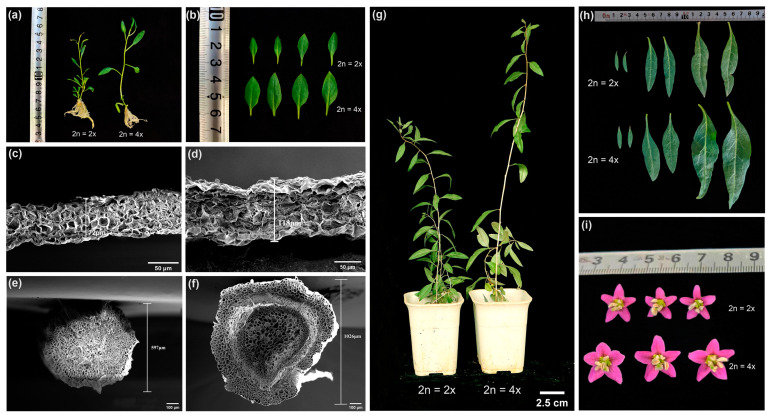
Morphological characteristics of *L. chinense* at different ploidy levels: (**a**) comparison of diploid (left) and tetraploid (right) plant height at 30 days; (**b**) comparison of growth differences between diploid and tetraploid leaves at leaf position 3–6; (**c**) diploid leaf cross-section under scanning electron microscopy; (**d**) tetraploid leaf cross-section under scanning electron microscopy; (**e**) diploid stem segment cross-section under scanning electron microscopy; (**f**) tetraploid stem segment cross-section under scanning electron microscopy; (**g**) comparison of diploid (**left**) and tetraploid (**right**) plant height at 120 days; (**h**) differences in growth of lower, middle, and upper leaves of diploids and tetraploids at 120 days of age; (**i**) morphological differences between diploid and tetraploid flower buds.

**Figure 5 plants-13-00439-f005:**
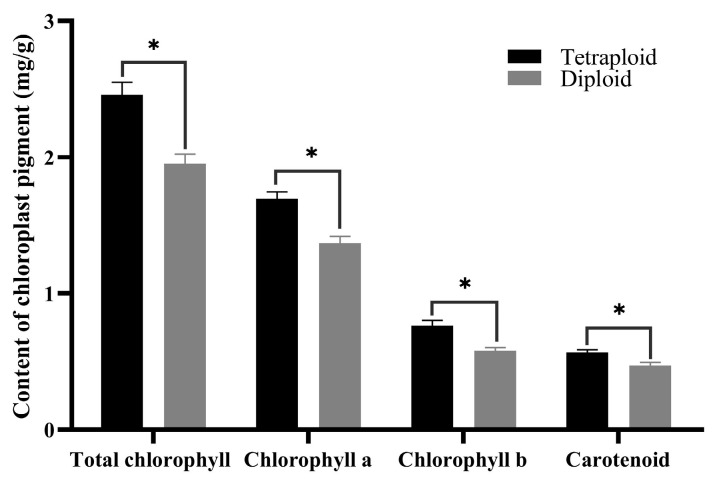
Chloroplast pigment content in diploid and tetraploid leaves. * for 0.01 < *p* ≤ 0.05.

**Figure 6 plants-13-00439-f006:**
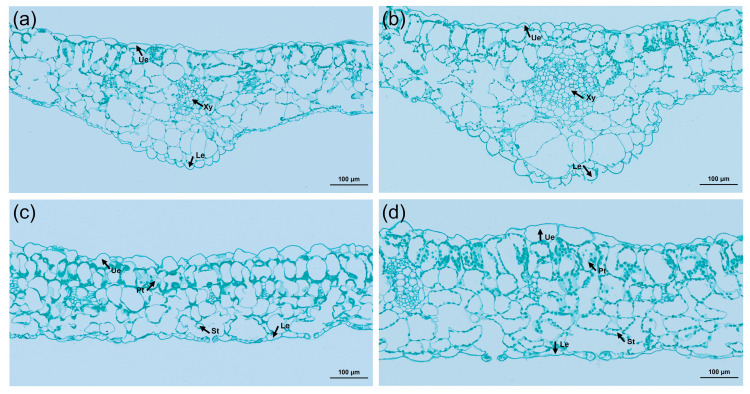
Cross-sectional anatomical characteristics of diploid and tetraploid leaves of *L. chinense*: (**a**) anatomical structure of diploid leaf vein; (**b**) anatomical structure of tetraploid leaf vein; (**c**) anatomy of the diploid leaf; (**d**) anatomy of the tetraploid leaf. Ue, upper epidermis; Le, lower epidermis; Xy, xylem; Pt, palisade tissue; St, spongy tissue.

**Figure 7 plants-13-00439-f007:**
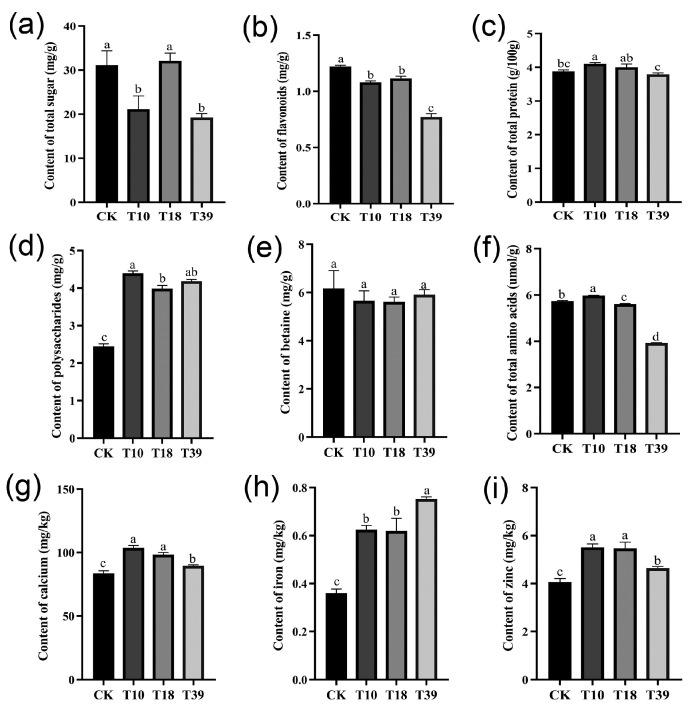
Comparison of differences in nutrient contents in leaves of diploid (CK) and tetraploid (T10, T18, and T39): (**a**) total sugar; (**b**) flavonoids; (**c**) total protein; (**d**) polysaccharides; (**e**) betaine; (**f**) total amino acids; (**g**) calcium; (**h**) iron; (**i**) zinc. The different letters in the same figure indicate significant differences based on Duncan’s multiple range test at the 0.05 probability level.

**Figure 8 plants-13-00439-f008:**
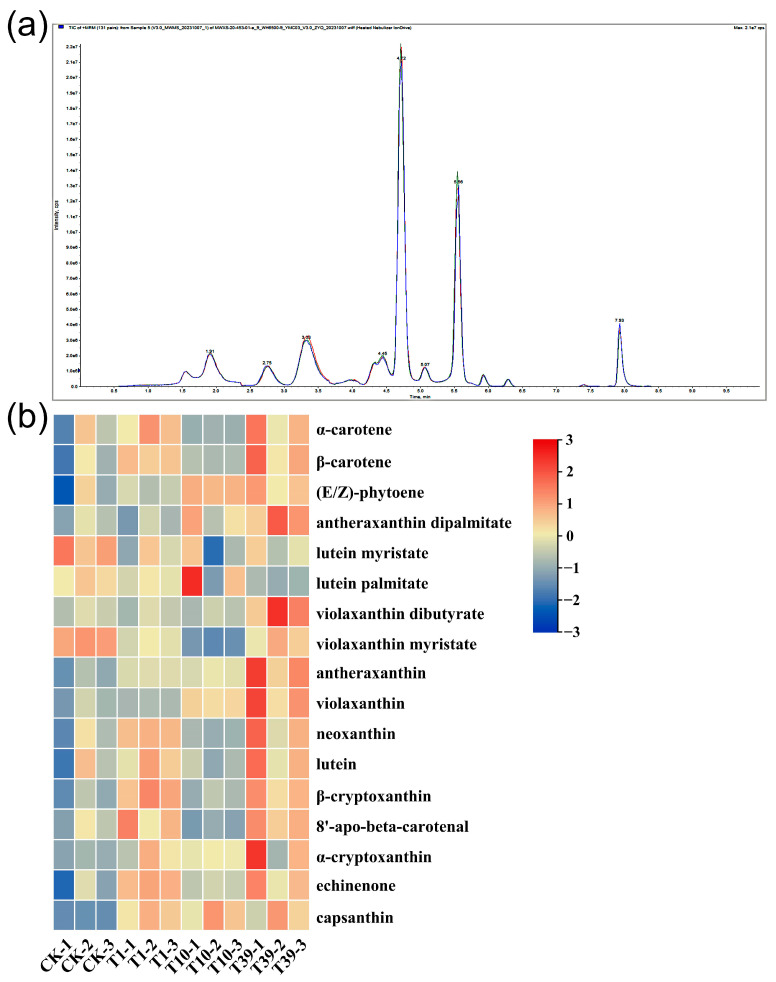
A heat map and overlay of the total ion current (TIC) of carotenoid metabolites in leaves of three tetraploid strains and the control: (**a**) TIC for mass spectrometry detection of QC samples; (**b**) A heat map of all carotenoid metabolites. Processing of data through the row scale, the high and low abundances of metabolites were indicated by red and blue, respectively.

**Table 1 plants-13-00439-t001:** Effects of 6-BA and NAA concentrations and ratios on shoot regeneration from leaf explants of *L. chinense*.

Treatment Number	6-BA (mg L^–1^)	NAA (mg L^–1^)	Regeneration Frequency (%)	Shoots per Explant (No.)
1	0.5	0.05	70.6 ± 0.9 d	2.96 ± 0.12 bcd
2	0.5	0.1	76.7 ± 2.1 c	3.21 ± 0.17 b
3	0.5	0.5	51.1 ± 1.5 g	2.01 ± 0.1 f
4	1	0.1	32.8 ± 1.3 i	1.99 ± 0.22 f
5	1	0.2	39.4 ± 1.2 h	2.15 ± 0.07 f
6	1	1	13.3 ± 1 j	1.29 ± 0.08 g
7	1.5	0.15	0 ± 0 k	0 ± 0 h
8	1.5	0.3	4.4 ± 0.7 k	1.25 ± 0.14 g
9	1.5	1.5	0 ± 0 k	0 ± 0 h
10	0.2	0.02	55.6 ± 1.8 fg	2.61 ± 0.2 de
11	0.2	0.04	63.9 ± 0.9 e	3.11 ± 0.13 bc
12	0.3	0.03	86.7 ± 1.5 b	3.92 ± 0.08 a
13	0.3	0.06	100 ± 0 a	4.26 ± 0.09 a
14	0.7	0.07	58.9 ± 1.8 ef	2.56 ± 0.12 e
15	0.7	0.14	63.3 ± 1 e	2.79 ± 0.05 cde

The data represent the mean ± SE of three replicates. The different letters in the same column data indicate significant differences based on Duncan’s multiple range test at the 0.05 probability level.

**Table 2 plants-13-00439-t002:** Effects of preculture time, colchicine concentration, and exposure time on survival rate and tetraploid induction frequency of *L. chinense*.

Treatment Number	Preculture Duration (Days)	Colchicine Concentration (mg L^–1^)	Exposure Time (h)	Survival Rate (%) ^a^	No. of Shoots Regenerate ^b^	No. of Tetraploid ^c^	Tetraploid Induction Rate (%)
1	10	40	24	96.1 ± 0.56	87	3	3.5
2			48	90.4 ± 0.98	65	3	4.6
3		50	24	91.5 ± 1.85	71	8	11.2
4			48	73 ± 2.06	44	3	6.9
5		60	24	69.3 ± 1.61	21	1	4.8
6			48	38.9 ± 1.7	7	0	0
7	12	40	24	90.4 ± 1.96	79	5	6.4
8			48	85.6 ± 1.28	62	3	4.9
9		50	24	87.8 ± 1.92	66	12	18.2
10			48	67.4 ± 0.98	40	7	17.1
11		60	24	60.7 ± 1.85	18	2	11
12			48	32.2 ± 2.94	5	1	16.7
13	14	40	24	88.5 ± 2.06	72	4	5.5
14			48	80.4 ± 1.34	53	4	7.5
15		50	24	83 ± 2.67	61	9	14.8
16			48	63.7 ± 1.96	37	5	13.6
17		60	24	42.6 ± 3.65	8	0	0
18			48	24.1 ± 3.29	3	0	0

^a^ Survival rate indicates the percentage of surviving explants relative to the total explants. Data represent the mean ± SE of three replicates. ^b,c^ Data represents the sum of three replicates.

**Table 3 plants-13-00439-t003:** Effect of ploidy level on stomatal and chloroplast characteristics of *L. chinense*.

Ploidy	Stomata Length (µm)	Stomata Width (µm)	Stomatal Density (N/mm^2^)	Chloroplasts Number (N/Stoma)
Diploids	28.13 ± 0.58 a	26.07 ± 0.48 a	157.9 ± 2.70 a	16.90 ± 0.44 a
Tetraploids	43.08 ± 0.67 b	35.33 ± 0.36 b	92.0 ± 3.26 b	26.73 ± 0.49 b

Data values are means ± SE. Lowercase letters indicate significant differences at the 0.05 probability level for the independent samples *t*-test.

**Table 4 plants-13-00439-t004:** Comparison of cytological traits of tetraploid and diploid leaves of *L. chinense*.

Characteristics	Diploid	Tetraploid	Significance
Leaf vein thickness (µm)	323.63 ± 18.92	466.12 ± 29.43	**
Upper epidermal thickness (µm)	16.17 ± 0.65	20.42 ± 0.64	****
Lower epidermal thickness (µm)	9.27 ± 0.20	12.87 ± 0.34	****
Palisade tissue thickness (µm)	41.16 ± 0.62	55.80 ± 1.03	****
Sponge tissue thickness (µm)	101.69 ± 0.86	123.16 ± 1.89	****
The ratio of palisade tissue to sponge tissue	0.41 ± 0.01	0.45 ± 0.01	***
Xylem cell diameter (µm)	10.86 ± 0.43	15.71 ± 0.74	****

Data values are means ± SE. ** for 0.001 < *p* ≤ 0.01, *** for 0.0001 < *p* ≤ 0.001, and **** for *p* ≤ 0.0001.

**Table 5 plants-13-00439-t005:** Carotenoids composition and content in tetraploids and diploids.

Composition	CK	T1	T10	T39
*α*-carotene	4.68 ± 1.29 ab	7.07 ± 0.66 a	3.95 ± 0.03 b	7.37 ± 0.93 a
*β*-carotene	241.16 ± 20.08 b	292.42 ± 2.43 a	247.87 ± 1.39 b	306.85 ± 17.09 a
(E/Z)-phytoene	2.43 ± 0.51 b	2.77 ± 0.08 ab	3.57 ± 0.03 a	3.44 ± 0.19 a
antheraxanthin dipalmitate lutein myristate	0.24 ± 0.01 b 0.10 ± 0.01 a	0.23 ± 0.01 b 0.07 ± 0.01 ab	0.27 ± 0.02 ab 0.06 ± 0.01 b	0.31 ± 0.02 a 0.07 ± 0.01 ab
lutein palmitate	0.18 ± 0.01 a	0.16 ± 0.01 a	0.20 ± 0.06 a	0.12 ± 0.00 a
violaxanthin dibutyrate	0.05 ± 0.00 b	0.04 ± 0.00 b	0.04 ± 0.00 b	0.09 ± 0.01 a
violaxanthin myristate	0.25 ± 0.00 a	0.18 ± 0.01 c	0.09 ± 0.00 d	0.21 ± 0.02 b
antheraxanthin	1.16 ± 0.08 b	1.44 ± 0.01 b	1.46 ± 0.02 b	1.94 ± 0.18 a
violaxanthin	120.11 ± 1.95 b	120.57 ± 0.19 b	127.96 ± 0.29 a	133.87 ± 3.82 a
neoxanthin	136.19 ± 9.09 b	161.35 ± 0.82 a	132.99 ± 0.89 b	162.69 ± 10.26 a
lutein	1708.60 ± 135.41 a	1910.82 ± 60.98 a	1684.59 ± 35.04 a	1973.82 ± 95.78 a
β-cryptoxanthin	2.02 ± 0.24 b	3.69 ± 0.18 a	2.25 ± 0.11 b	3.51 ± 0.24 a
8′-apo-beta-carotenal	0.02 ± 0.00 b	0.03 ± 0.00 a	0.02 ± 0.00 b	0.03 ± 0.00 a
α-cryptoxanthin	0.09 ± 0.01 a	0.17 ± 0.03 a	0.16 ± 0.00 a	0.21 ± 0.06 a
echinenone	0.04 ± 0.01 c	0.07 ± 0.00 a	0.05 ± 0.00 bc	0.06 ± 0.01 ab
capsanthin	0.07 ± 0.00 b	0.07 ± 0.00 a	0.07 ± 0.00 a	0.07 ± 0.00 a

The data represent the mean (µg/g) ± SE of three replicates. The different letters in the same column data indicate significant differences based on Duncan’s multiple range test at the 0.05 probability level.

## Data Availability

Data is contained within the article or Appendix A.

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
