# Peer review of "Chromosome Doubling Enhances Biomass and Carotenoid Content in Lycium chinense"

_plants, 2024, doi:10.3390/plants13030439_

Round 1

Reviewer 1 Report

Comments and Suggestions for Authors

need some edit as per comments

Comments on the Quality of English Language

Reviewer 2 Report

Comments and Suggestions for Authors

NA

Comments on the Quality of English Language

The study is interesting and may be useful to improve the L. chinense. I am suggesting here few revisions which may improve the quality and clarity of the paper. 

1. Rewrite the abstract by including the quantitative data of the traits especially in the lines 16 to 20. 

2.  In introduction, it was mentioned that the demand has been increasing but reference or data has been shown. I will suggest to include the data. 

In results: 

Line 79: leaf cuts were transfered to the medium, which medium?

line 86: in Figure 1 title, please change the word 'inoculation' and better use appropriate word

Line 90: elaborate NAA & 6-BA as these words introduced first time

Line 100: In 1st column of table 1, use number or serial after the 'treatment'

Line 100-134: P-value has been used but no data or analysis found anywhere in the table? you can add the test-results or remove the unsupported data or information.

Discussion: 

Discussion must be re-write in a way by explaining the findings and the due reasons, as well as inclusion of literature supports and future perspectives. 

Current discussion is lacked of the explanation of the quantitative findings.     

Reviewer 3 Report

Comments and Suggestions for Authors

The authors has successfully produced tetraploid lines of Lycium and demonstrated that the concentration of carotenoid has been increased compared to diploid lines, and other morphological traits as well has increased in size, which is expected from polyploidization. The authors state in the abstract "The findings suggest that the generated tetraploids harbor significant potential for further exploitation" which may be correct but the study should go further to demonstrate the claim in order to have impact for "Plants". This method development paper is more pertinent for dedicated journals in vitro culture. The experimental work and presentation is well done. I think there are far too many citation as the experimental work setting up a protocol for producing tetraploid lines by means of known technologies.

Round 2

Reviewer 3 Report

Comments and Suggestions for Authors

Now the abstract better explain what has been achieved, and reading the Introduction it becomes clear that the focus for plant improvement is the leaves and not the berries, for increasing the concentration of nutritional important compounds. However, what was the concentration of carotenoids in the tetraploid berries, which by-the-way is widely used in breakfast foods and energy-bars. Will the field production be based on vegetatively propagated plant lines or will they undergo a life cycle and seed produced for the next generation?

line 35 change ".Wolfberry contains...." to " Wolfberry (Lycium barbarum) also known as goji berry, contains...."

Ken Qin has been added as author and listed in the author contribution, is that approved by the journal?
